# Solid-State Formation of a Potential Melphalan Delivery Nanosystem Based on β-Cyclodextrin and Silver Nanoparticles

**DOI:** 10.3390/ijms24043990

**Published:** 2023-02-16

**Authors:** Rodrigo Sierpe, Orlando Donoso-González, Erika Lang, Michael Noyong, Ulrich Simon, Marcelo J. Kogan, Nicolás Yutronic

**Affiliations:** 1Departamento de Química, Facultad de Ciencias, Universidad de Chile, Las Palmeras #3425, Ñuñoa, Santiago 7800003, Chile; 2Departamento de Química Farmacológica y Toxicológica, Facultad de Ciencias Químicas y Farmacéuticas, Universidad de Chile, Sergio Livingstone #1007, Independencia, Santiago 8380492, Chile; 3Departamento de Química, Facultad de Ciencias Naturales, Matemática y del Medio Ambiente, Universidad Tecnológica Metropolitana (UTEM), Las Palmeras 3360, Ñuñoa, Santiago 7800003, Chile; 4Advanced Center for Chronic Diseases (ACCDiS), Sergio Livingstone #1007, Independencia, Santiago 8380492, Chile; 5Institute of Inorganic Chemistry, RWTH Aachen University, Landoltweg 1a, D-52074 Aachen, Germany

**Keywords:** melphalan, cyclodextrin, silver nanoparticle, inclusion complex, drug delivery, solid-state formation, nanomaterial, sputtering

## Abstract

Melphalan (Mel) is an antineoplastic widely used in cancer and other diseases. Its low solubility, rapid hydrolysis, and non-specificity limit its therapeutic performance. To overcome these disadvantages, Mel was included in β-cyclodextrin (βCD), which is a macromolecule that increases its aqueous solubility and stability, among other properties. Additionally, the βCD–Mel complex has been used as a substrate to deposit silver nanoparticles (AgNPs) through magnetron sputtering, forming the βCD–Mel–AgNPs crystalline system. Different techniques showed that the complex (stoichiometric ratio 1:1) has a loading capacity of 27%, an association constant of 625 M^−1^, and a degree of solubilization of 0.034. Added to this, Mel is partially included, exposing the NH_2_ and COOH groups that stabilize AgNPs in the solid state, with an average size of 15 ± 3 nm. Its dissolution results in a colloidal solution of AgNPs covered by multiple layers of the βCD–Mel complex, with a hydrodynamic diameter of 116 nm, a PDI of 0.4, and a surface charge of 19 mV. The in vitro permeability assays show that the effective permeability of Mel increased using βCD and AgNPs. This novel nanosystem based on βCD and AgNPs is a promising candidate as a Mel nanocarrier for cancer therapy.

## 1. Introduction

Melphalan (Mel) is a bifunctional alkylating agent widely used at the clinical level in the treatment of multiple types of cancer and other diseases [1,2,3,4]. The effectiveness of killing cancer cells of this drug is based on the ability to react extensively with DNA, RNA, and proteins to form interstrand DNA crosslinks, inducing multiple kinds of molecular lesions [5,6,7]. Nevertheless, Mel presents serious therapeutic disadvantages related to its non-specificity, the resistance of tumor cells, and the need for increasingly higher doses during treatment, generating adverse effects that include leukopenia, mucositis, and diarrhea [8,9]. In this sense, the administration of Mel could be optimized using nanotechnology [10,11,12,13,14,15,16,17]. Specifically, silver nanoparticles (AgNPs) have been used for the loading and transport of antitumor drugs, showing a synergistic effect with these drugs and a site-specific action, which would reduce the administered dose and even reduce the toxicity in healthy cells [18,19,20]. This research field is controversial; therefore, it is necessary to expand on this point.

The properties of AgNPs intrinsically depend on their size, shape, and surface coating; therefore, if properly designed, they can be used in cancer therapy [21,22]. AgNPs accumulate specifically at tumor sites due to the enhanced permeation and retention effect (called EPR effect); this allows drug targeting if it acts as a nanocarrier [23,24]. In addition to the therapeutic effect of the material itself, which has demonstrated antiproliferative activity against tumor cells of different types [25,26]. On the other hand, due to their excellent optoelectronic properties, they exhibit a localized surface plasmon resonance that allows for photothermal therapy [21,27]. This consists of the conversion of the irradiated energy on these nanoparticles into local heat, which triggers hyperthermia and release of the bound therapeutic material [28,29,30]; this is especially relevant for the treatment of tumors, for example, one generated by skin cancer, where the therapy must operate on the most external layers of the tissue [31,32,33]. Likewise, the therapy based on Mel laser irradiation has also been studied by some authors with favorable results on organ transplantation [34,35,36].

Although Mel vectorization on AgNPs could optimize their transport and synergize at the therapeutic level, solubility and stability in water remain an issue. Current formulations must be prepared in situ as hydrolysis decomposes Mel into a non-active molecule completely within 8 h at 37 °C [37]. In addition, the poor aqueous solubility, less than 1 mg/mL, severely limits the dose administered [3,4,9]. In this sense, a promising approach to overcome these limitations has been the loading of this type of drug into β-cyclodextrin (βCD), forming inclusion complexes [38,39,40]. βCD is a cyclic oligosaccharide macromolecule, approved by FDA (Food and Drug Administration), composed of 7 glucose units polymerized that have a shape similar to a truncated cone. The use of βCD has a wide range of applications ranging from pharmaceutical formulations, food products, and cosmetics, among others. It has a hydrophobic interior and a hydrophilic exterior that allows non-polar molecules of certain dimensions to be included in its interior, forming a stable complex, preventing hydrolysis, and increasing solubility [41,42]. In general, the toxicity of cyclodextrins depends on the route of administration; by the oral route, they are practically non-toxic; by the parenteral route, they are considered safe; however, increasingly higher doses and their accumulation could be harmful, although if it is administered intravenously, they rapidly disappear from the systemic circulation and are excreted via the kidneys [39,43]. Notably, the inclusion of Mel in cyclodextrin derivatives has been studied, with encouraging results regarding toxicity, stability, and solubility of the unloaded drug [44,45,46]. In recent years, a research line on cyclodextrin–drug complexes combined with plasmonic nanoparticles has been developed with promising results. The crystals formed by the drug included in the macromolecule acquire an arrangement that allows the stabilization of gold nanoparticles in the solid state, creating new nanosystems with optimal release and performance parameters [13,14,15,17,47]. A solid formulation containing the drug is a strategy that could optimize its physicochemical properties, such as chemical and physical stability, solubility, and bioavailability, among others. In addition, it allows the development of formulations for different routes of administration, which is especially relevant for Mel.

In this work, two tools were combined synergistically aimed at optimizing the use of Mel in chemotherapy. First, the βCD matrix was used to form the βCD–Mel complex in a solid state. Subsequently, AgNPs were formed and immobilized directly on these crystals by the physical method of magnetron sputtering, thus avoiding the use of agents considered toxic for biomedical applications. Finally, the system was solubilized to obtain a colloidal solution of AgNPs covered by the complex. A schematic representation of the nanosystem formation process at each stage is shown in Figure 1. During its development, this new βCD–Mel–AgNPs crystalline system was characterized using techniques such as powder X-ray diffraction, one and two-dimensional NMR, UV-vis spectroscopy, and scanning electron microscopy. Then, the βCD–Mel crystals with AgNPs on their surface were dissolved, forming the new βCD–Mel–AgNPs colloidal nanosystem, which was characterized using dynamic light scattering, zeta potential, and transmission electron microscopy. The study of relevant pharmaceutical parameters, such as association constant, loading capacity, and nanosystem size, among others, were included. In addition, the effective permeability of Mel in βCD and with AgNPs was evaluated in an in vitro membrane model. Our research is the first report of Mel complexation in βCD and its interaction with AgNPs in a solid state and surfactant-free, where both components work to solve some therapeutic drawbacks of this drug and as a potential nanocarrier.

## 2. Results and Discussion

### 2.1. Formation of the Complex in Solid State

The diffraction patterns of the βCD, Mel, and the complex formed were obtained using PXRD (see Figure 2). The formation of the complex in the solid state causes the disappearance of the Mel peaks, especially in the region above 25° 2θ, maintaining a peak corresponding to βCD at 2θ angle of approximately 12°. The crystal packing arrangement of the complex was different from that of its pure species and represents a new crystalline phase, confirming the effective inclusion of the drug in the βCD matrix. Three intense peaks were observed at approximately 4°, 12° and 20° 2θ in the βCD–Mel pattern. It has been reported that the formation of complexes with a P2_1_-type structure for βCD matrices hosting aromatic rings at 1:1 (host–guest) molar ratios also exhibit three intense peaks at equal locations. In addition, the geometrical arrangement and thus the respective diffraction pattern varies according to the type of molecular structure of the aromatic guest, according to the region partially exposed to the outside of the βCD cavity and to the water molecules contributing to the new crystalline packing [48,49,50]. Regarding the physical mixture of pure components, (Figure 2d), it was observed that its diffractogram corresponds to a superposition between the traces of the pure species.

### 2.2. Stability of the Complex in Solution

^1^H-NMR was conducted to study the conformation and stoichiometry of βCD–Mel and to evaluate the stability of Mel included in the complex. The latter was chosen because, in therapy, Mel has poor solubility in water and is unstable on reconstitution and dilution [44]. In 1978, Chang et al. proposed that the hydrolysis of Mel occurs in three phases where chloroethyls, one by one, are transformed into hydroxyethyls by a three-component cyclic intermediate mechanism with the tertiary amine. This process is completed after 8 h [37].

Once the complex was formed in the solid state, it was dissolved, ^1^H-NMR spectrum of βCD–Mel in solution was registered and then compared with the free βCD and Mel spectra (see Figure 3).

Table 1 shows the chemical shifts of βCD–Mel compared to its pure species. The shifts of the 2′/6′, 3′/5′ protons of the ring and the 1″ and 2″ protons of the drug chain towards lower fields demonstrate interaction with the CD matrix; while the chemical shifts of the H′-3 a and b protons of Mel suggest interactions with the OH groups present on the surface of the βCD cavity. The internal H-3, H-5, and H-6 matrix protons demonstrate that the drug remains included. The OH-6 groups move to higher fields due to the proximity of electronegative groups such as COOH and NH_2_ that are close to one of the CD openings. Considering the changes observed in Mel upon hydrolyzing and the signals observed in the spectrum, it is confirmed that through the formation of the complex, the drug can remain stable within the matrix, considerably increasing its applicability in therapy.

Integrating the proton signals of βCD and Mel in the ^1^H-NMR spectrum of the complex allowed for the determination of the host:guest stoichiometric ratio in solution, according to a methodology used for different CD complexes [51,52]. The integrations of the H-1 proton of the βCD with H′-2′/6′ and H′-3/5′ protons of the aromatic ring of Mel were compared. The results show that the complex was formed in a 1:1 stoichiometric ratio, being relevant for potential applications (more details are given in the Appendix A).

### 2.3. Drug Loading and Complexation Efficiency

In pharmaceutical concepts, knowing the stoichiometric ratio allows us to obtain important parameters for potential formulation production [13,14,15,16,53]. The 1:1 drug/βCD molar ratio means that the 0.75 g of cyclodextrin can load 0.20 g of drug, so the loading capacity of the matrix was 27%. The solubility of βCD in water is 18.5 g/L [54], so to completely dissolve the inclusion complex, about 40.5 mL is needed, where 200 mg of the drug would be found. This means that the formation of the complex increases the practical solubility of Mel almost five times.

On the other hand, the phase solubility method allows knowing parameters such as the degree of solubilization, association constant, and complexation efficiency [53] (further details of the procedure and figures are provided in the Appendix A). Linear increases in the aqueous solubilities of Mel versus βCD used were observed, with a 0.034 degree of solubilization. In general, the recommendable *K*_a_ values of drug:CD complexes are between 50 and 2000 M^−1^, because at values below 50 M^−1^, the pharmaceutical formulation may be limited by its low stability, not favoring a controlled release, e.g., an early liberation into the blood and not into the desired tissue. On the other hand, at high *K*_a_ values (>2000 M^−1^), the ability to release the drug by gradient or by stimulus can be affected by the high degree of inclusion of the drug in the βCD cavity, excessively delaying its pharmacokinetics, among other possible therapeutic drawbacks [55,56]. The *K_a_* of βCD–Mel was 625 M^−1^, being located in an adequate range, with stable host–guest interactions, which prevent its early release and could allow its delivery to the site of action. Furthermore, the complexation efficiency can be used to probe the feasibility of using cyclodextrins in the formulation of drugs. In the βCD–Mel system, the complexation efficiency value was 0.035, being suitable for a potential formulation based on this complex.

### 2.4. Molecular Interaction and Geometry of the Inclusion Complex

Whereas the formation, stability, and loading parameters of Mel on the complex were in the desired range, for the formation of the nanovehicle by stabilizing AgNPs directly on its surface using βCD, it was necessary to know the availability of adequate exposed functional groups. The rotational overhauser enhancement spectroscopy (ROESY) technique enables observation of interactions between hydrogen nuclei of non-chemically bonded structures separated by a maximum distance of 5 Å; therefore, it is possible to determine in solution the geometrical structure of supramolecular complexes accurately [51,52]. The ROESY spectra of the βCD–Mel complex are shown in Figure 4a,b. A model showing the proposed geometry of βCD–Mel is shown in Figure 4c (full ROESY spectra are shown in the Appendix A).

The analysis reveals that the H′-2′/6′ and H′-3′/5′ protons, corresponding to the aromatic ring of Mel, exhibit a correlation with the H-3 and H-5 protons of βCD (Figure 4a). This indicates that the aromatic ring is located in the wider region of the matrix. The exact orientation of the drug can be elucidated from the cross-peaks observed between the internal protons of βCD with the H′-3a proton of Mel (Figure 4b). This reveals that Mel is exposed toward the major opening, and thus, the chloroethyl chains are oriented toward the minor opening of the matrix. These results explain the chemical shifts of Mel in the ^1^H-NMR spectrum, where shifts towards lower fields of the OH-6 and the H′-1″ and H′-2″ protons were observed. This confirms that it is the interaction of the primary hydroxyl groups of βCD with the chloroethyl chains of Mel that causes this effect. Accordingly, by the dimensions of both molecules, the NMR and ROESY result strongly suggests that the COOH and NH_2_ functional groups of Mel are oriented outside the cavity of the matrix, thus making it a suitable candidate to stabilize AgNPs.

### 2.5. Deposition of Silver Nanoparticles on the βcyclodextrin–Melphalan Complex

AgNPs were obtained and deposited using the physical method of magnetron sputtering in vacuum. The technique consists of the detachment of atoms from a thin metallic film, which agglomerate and stabilize on a substrate, reaching a nanometric size [13,14,15,57,58]. As we have confirmed, Mel was partially included within βCD, exposing the amine and carboxylic acid functional groups towards one of its openings. This arrangement allowed the solid-state complexes to act as substrates and the crystalline faces that expose these functional groups to interact specifically with silver atoms allowing its accumulation, forming, and stabilizing nanoparticles. Figure 5a shows a schematic representation of this process. The new βCD–Mel–AgNPs crystalline system was characterized using UV-vis diffuse reflectance spectroscopy, SEM, EDX, and FE-SEM. Figure 5b corresponds to the absorbance spectra of AgNPs deposited on the βCD–Mel complexes. The maximum absorbance peak in the SPR of the nanoparticles was located at 450 nm (see the UV-vis spectra of Mel in solid state in the Appendix A). 

It has been reported that AgNPs with an average diameter of 10 nm show an SPR with a maximum absorbance peak at 400 nm [59]. There are reports of AgNPs formation on different substrates using this method. In these studies, it is argued that the bathochromic shift of the band above 400 nm is probably due to an increase in the size of the formed nanoparticles (above 10 nm), to the change in the dielectric environment or to an interparticle plasmon coupling [57,60,61,62].

Figure 6 corresponds to SEM images showing directly the nanoparticles deposited on crystals of the complex, together with an elemental analysis performed on a section of the crystal using EDX (see more FE-SEM images and EDX table information in Appendix A).

The phenomenon of the inclusion of molecules on βCD generates crystalline arrangements different from that of their pure species. In this case, it was observed that the crystalline morphology of the βCD–Mel complex is different with respect to the pure compounds βCD and Mel (see SEM images information in Appendix A). 

This corroborates what was previously discussed in the analysis of the samples by XRD. Figure 6a,b shows a high concentration of AgNPs of homogeneous size, preferentially accumulated on one side of the crystal, where NH_2_ and COOH functional groups of Mel are exposed, allowing its stabilization and confirming that the phenomenon of aggregation is widely suppressed. The EDX spectra obtained from the surface area analyzed show the atomic and weight elemental composition of the sample (see Figure 6c). The main components were carbon and oxygen; chlorine is also included, coming from the drug, and silver from the nanoparticles that remain stable in the solid state on the surface of the crystals. 

### 2.6. Formation of βcyclodextrin–Melphalan–Silver Nanoparticles Nanosystem

The βCD–Mel crystals with deposited AgNPs were solubilized in water, promoting the formation of a colloidal solution of AgNPs stabilized by the βCD–Mel complex, as schematized in Figure 7a. The new ꞵCD–Mel–AgNPs colloidal nanosystem was characterized by DLS, zeta potential, and TEM. The results are summarized in Table 2.

The solvation sphere diameter of the nanosystem was 116 nm with a polydispersity of 0.4, characteristic of ꞵCD-based nanosystems [12,16,17,63,64,65]. Therefore, the dissolution process causes the metal nanoparticles to be surrounded by multiple layers of the complex, as has been previously reported for analogous nanosystems using gold nanoparticles [13,14,15]. In addition, the drug remains partially included in the cavity of the βCD, and its functional groups interact with the surface of Ag atoms. The above was confirmed by the positive surface charge of the ꞵCD–Mel–AgNPs colloidal nanosystem, which was attributed to the NH_2_ groups of Mel [66,67,68]. The size of spherical AgNPs was determined using micrographs obtained by TEM (see Figure 7b). A high concentration of AgNPs very close to each other was observed, homogeneous in size and shape, mostly in the range of 10 to 20 nm. Through the analysis of the respective histogram, an average diameter of 15 ± 3 nm was reported. The difference between the TEM size and the hydrodynamic diameter is explained by the multiple layers of ꞵCD–Mel, which cover the nanoparticles and maintain the stability of the colloid. Interestingly, for metal nanoparticles above 5 nm, the size is not a critical parameter in the toxicity profile of a nanosystem, since it has no disruptive effects on healthy cells and can be rapidly excreted by the body [24,69]. Once they accumulate in fenestrated tissue, such as tumors, the expected long-term toxicity of AgNPs is due to the oxidative release of Ag^+^ that could act synergistically for potential treatment [26]. Accordingly, the average size of the nanoparticles and the hydrodynamic diameter of the βCD–Mel–AgNPs colloidal nanosystem new open the way to possible applications in therapy, in addition to being a stable nanocarrier for Mel. In this sense, one of the relevant tests to validate this new system as a nanocarrier in solution is to evaluate its permeability in membranes. For this, studies using the PAMPA method were performed.

### 2.7. Permeability Assays on Artificial Membranes

The PAMPA method is used to predict the passive permeability of biologically relevant species, such as drugs, across the membrane. The assay is performed on a donor and acceptor plate, in which the ability of a compound to diffuse between the two plates separated by a phosphatidylcholine filter is measured, which acts as a lipid membrane and simulates the lipid bilayer of different types of cells. Since artificial membranes do not contain active transport systems, only passive diffusion can be studied [70,71]. Cyclodextrins can enhance the permeability of drugs through an artificial membrane under certain conditions. However, cyclodextrin-based complexes increase the total amount of drug in the solution and the concentration gradient. The molecule excess can decrease penetration, especially lipophilic drugs that could permeate the membrane. Other factors such as agitation speed, pH, type of cyclodextrin, or the association constant of each inclusion complex are variables to be considered [72,73].

Figure 8 shows the effective permeability of Mel, βCD–Mel complex, and βCD–Mel–AgNPs colloidal nanosystem analyzed by PAMPA. The controls used behaved as expected, indicating that the membranes were well constructed. In the case of Mel, as part of the complex, its effective permeability increased with respect to Mel alone, which was consistent with the role of βCD, maintaining drug stability and, in turn, altering membrane properties [74,75]. In addition, Mel, in the presence of AgNPs, also increases its effective permeability with respect to Mel alone, probably because the colloid is surrounded by multiple layers of the βCD–Mel complex [76,77]; however, it does not exceed the permeability of the complex. This is explained by the interaction of the NH_2_ functional group of Mel with the surface of the AgNPs, decreasing their migration into the membrane. The foregoing suggests that AgNPs maintain the stability and composition of the nanosystem under the experimental conditions studied, which qualifies the system as a promising candidate for site-directed drug transport [18,26,78,79,80] since it would avoid its non-specificity or early action in healthy cells during systemic circulation. On the other hand, it conditions the nanocarrier to release the drug through an activation with external stimuli to the biological environment, such as laser irradiation [11,15,17,19,47,81].

## 3. Materials and Methods

### 3.1. Reagents and Solvents

In the preparation of the complex, the following reagents were used: βCD hydrate (98%, 1134.98 g/mol), Mel (>95%, 305.2 g/mol), Dimethyl sulfoxide (DMSO)-d_6_ (99.99% D, 84.17 g/mol), Phosphatidylcholine (99%), Dodecane (99%, 170.33 g/mol), Phosphate buffered saline (PBS) tablets, Thiopental (99%, 242.34 g/mol), and Evans Blue (75%, 960.81) provided by Sigma-Aldrich (Saint Louis, MO, USA). As solvents, ethanol p.a. and water for chromatography LiChrosolv^®^ brand Merck (Darmstadt, Germany) were employed. To obtain the AgNPs via magnetron sputtering, a silver foil of high purity (99.9%) was used as a precursor.

### 3.2. Synthesis of the βCD–Mel Complex

To synthesize the βCD–Mel complex, the saturated solutions method was employed [13,14,15,17,82,83]. In total, 0.2025 g of Mel were dissolved in ethanol and then added to a solution of βCD (0.7530 g dissolved in water) at 4 °C with gentle and constant stirring, allowing the temperature to rise slowly up to room temperature. The new solution was kept motionless under a hood for one week until the evaporation of the solvents. Finally, small crystals precipitated at the bottom of the crystallizer were extracted and washed with a 50% *v*/*v* solution (ethanol/water) at 4 °C and vacuum filtered for one hour in a kitasate with a Büchner funnel equipped with a double layer of filter paper, to remove the excesses of cyclodextrin, drug or solvents that moisten the complex. The crystals were then pulverized and dried in a vacuum line to remove the water or ethanol, which may remain occluded in the powder, then stored in amber vials with a Teflon seal. The new βCD–Mel complex was characterized by powder X-ray diffraction (PXRD), NMR of one (^1^H) and two (ROESY) dimensions, UV-vis spectroscopy for calculating the loading capacity, degree of solubilization, complexation efficiency, and association constant. The complex was also used for a permeability study called PAMPA (parallel artificial membrane permeability assay).

### 3.3. Formation of Silver Nanoparticles

AgNPs were obtained using magnetron sputtering under a high vacuum, depositing them onto crystalline powder of the βCD–Mel complex acting as a substrate [57,84,85,86]. In total, approximately 20 mg of this complex was dispersed on a glass slide, which was placed inside the chamber of the equipment, together with a silver foil serving as a cathode. The chamber was evacuated to 0.5 mbar; then, Argon was injected together with a current of 25 mA to ionize the gas, which hit the metal foil, releasing silver atoms that afterward were deposited and stabilized on the complex. The accumulation of atoms on specific crystalline faces of βCD–Mel forms AgNPs in a process that lasts for 20 s. βCD–Mel–AgNPs was characterized by UV-vis spectroscopy, field emission scanning electron microscopy (FE-SEM), energy-dispersive X-ray spectroscopy (EDX), dynamic light scattering (DLS), zeta potential, transmission electronic microscopy (TEM), and PAMPA.

### 3.4. Powder X-ray Diffraction (PXRD)

The analysis of the complex and pure species βCD and Mel was performed using a Siemens D-5000 diffractometer with graphite-monochromated Cu K-α radiation at 40 kV and 30 mA with a wavelength of 1.540598 Å.

### 3.5. ^1^H-NMR and ROESY

Measurements for the complex and pure species βCD and Mel were performed at 300 K in DMSO-d_6_ (99.99%) on a Bruker Advance 400 MHz superconducting NMR spectrometer. All 2D NMR spectra were acquired using pulsed-field gradient-selected methods during a mixing time of 12 h. Additionally, the stoichiometry of the complex was evaluated using the integration of the signals of the protons of βCD and Mel in the spectrum of βCD–Mel.

### 3.6. Loading Capacity

The loading capacity of the βCD–Mel was calculated from the weights of βCD and drug obtained using Equation (1) [87].
(1)$$Loading\;capacity=\frac{Weight\;of\;drug\;in\;\beta CD}{Weight\;of\;\beta CD}\times 100$$

### 3.7. Phase Solubility Studies

Studies were performed following the Higuchi and Connors method [53]. The Beer–Lambert law was used to quantify Mel, considering its absorbance at 301 nm. First, known concentrations (C) of Mel were measured by UV-vis. From the UV-vis spectra, the absorbance maxima at 301 nm (A_max_) were extracted. The slope from the A_max_ vs. C graph, corresponded to the ε of the drug (For details, graphs, and tables, see Appendix A). Then, the βCD concentration versus the loaded Mel concentration (calculated by Beer–Lambert law) was plotted. The value of the slope of the graphs related to the amount of βCD added to the amount of solubilized drug, indicating the degree of solubilization. The degree of solubilization was used to calculate the association constant (*K_a_*) and complexation efficiency of the βCD–Mel system using Equations (2) and (3), respectively [16].
(2)$$K_{a}{}_{\left(1:1\right)}\;=\;\frac{Degree\;of\;solubilization}{\left[C_{o}\right]\left(1-Degree\;of\;solubilization\right)}$$
(3)$$Complexation\;efficiency=K_{a}{}_{\left(1:1\right)}\left[C_{o}\right]\;=\;\frac{Degree\;of\;solubilization}{\left(1-Degree\;of\;solubilization\right)}$$

[C_o_] corresponds to the concentration of the free drug in the absence of βCD.

### 3.8. UV-Vis Spectroscopy in Solid State

The AgNPs deposited onto βCD–Mel were characterized using UV-vis spectroscopy in a solid state [88,89,90]. First, the diffuse reflectance was measured using a Shimadzu UV 2450 spectrophotometer with barium sulfate as a baseline. In addition, the spectrum of the complex was used as a second baseline. Then, the absorbance of the AgNPs was determined using the Kubelka–Munk transformation. 

### 3.9. (Field Emission-) Scanning Electron Microscopy and Energy Dispersive Spectroscopy

SEM images were obtained using an LEO 1420VP equipment with an Oxford 7424 energy dispersive spectrometer coupled at an accelerating voltage of 25 kV. FE-SEM images were obtained using a Zeiss Leo Supra 35-VP at an accelerating voltage of 15 kV and 20 kV.

### 3.10. Dynamic Light Scattering and Zeta Potential Measurements

Dynamic light scattering and zeta potential measurements were performed at 25 °C using a Zetasizer model Nano ZS. The size distribution of the samples was determined from the results of the intensity distribution values using the cumulant method. The Smoluchowski approximation was used to calculate zeta potentials based on measured electrophoretic mobility.

### 3.11. Transmission Electron Microscopy

TEM images were obtained using a JEOL JEM 1200 EX II instrument. The samples with AgNPs were prepared by dispersing approximately 0.5 mg in 100 μL of isopropanol (30%). Then, 20 μL of the solution was deposited onto a copper grid with a continuous film of Formvar, the excess solution was removed, and the grid was dried. The acceleration voltage used was 80 kV. More than two thousand nanoparticles were counted to produce the size distribution histogram. The average diameter was obtained from the peak of the Gaussian fit performed on the graph.

### 3.12. Parallel Artificial Membrane Permeability Assay (PAMPA)

The *PAMPA* studies require two types of well, a donor and an acceptor (Transwell plates); in addition, the acceptor well has a semipermeable membrane of PVDF (polyvinylidene fluoride). On the PVDF membranes were deposited 4.0 μL of a solution of phosphatidylcholine in dodecane (20 mg/mL) for 5 min until complete evaporation of the organic solvent. Subsequently, 300 μL of PBS (pH = 7.4) was added. In the donor well were deposited 300 μL of each sample dissolved in PBS, which are: Mel, βCD–Mel, and βCD–Mel–AgNPs (all concentration data in Appendix A). As positive and negative controls, a solution of Thiopental (10 mg in 100 mL of PBS 10 mM) and a solution of Evans Blue (200 mg in 100 mL of PBS 10 mM) were used, respectively. After loading the donor well with the samples and the controls, and the acceptor well with PBS, they were assembled, covered, and sealed using parafilm. Each PAMPA was performed in triplicate (*n* = 3) for 24 h at 37 °C with constant agitation to 280 rpm.

The effective permeability (Pe) was determined by the following formula:(4)$$P_{e}=\frac{-218.3}{t}\;\cdot \;\log\left[1-\frac{2\;\cdot {\;C}_{A}\left(t\right)}{C_{D}\left(0\right)}\right]\;\cdot \;10^{-6}\;\mathrm{cm}/s$$

t = measurement time in hours. 

CA = concentration in acceptor plate at time t.

CD = concentration in donor plate at time zero.

To calculate the concentrations in the acceptor plate, the absorbances of the different plates were measured using a UV-vis spectrophotometer and the respective calibration curve of Mel in PBS [70,71].

## 4. Conclusions

In summary, a βCD–Mel crystalline solid complex was formed reproducibly in a 1:1 molar ratio. The hydrophobic cavity of βCD allowed the inclusion of Mel, which prevents hydrolysis and subsequent degradation of the drug. The exhaustive characterization of the complex confirmed that Mel partially exposes the -NH_2_ and COOH functional groups, allowing the crystals to act as a substrate for the stabilization of silver atoms, and the consequent formation of AgNPs using magnetron sputtering. In turn, the solubilization of the βCD–Mel–AgNPs crystalline system generates a colloidal solution, where the AgNPs are covered in multiple layers by the complex. The loading capacity, the stability of the drug, and the association constant of the complex, together with the size achieved of the nanoparticles and the effective permeability examined for βCD–Mel and βCD–Mel–AgNPs are promising results that validate this nanosystem as a new drug nanocarrier for Mel. Accordingly, the effective and stable Mel loading in this new nanosystem based on βCD and AgNPs shows potential as a novel alternative to traditional cancer therapeutics.

As a future perspective, evaluating the synergistic effect of the components of the nanosystem, together with studies of cytotoxicity and laser-triggered controlled release of Mel in in vitro and in vivo models, are considered necessary to position this nanosystem as a useful tool in drug delivery applied to cancer.

## Figures and Tables

**Figure 1 ijms-24-03990-f001:**
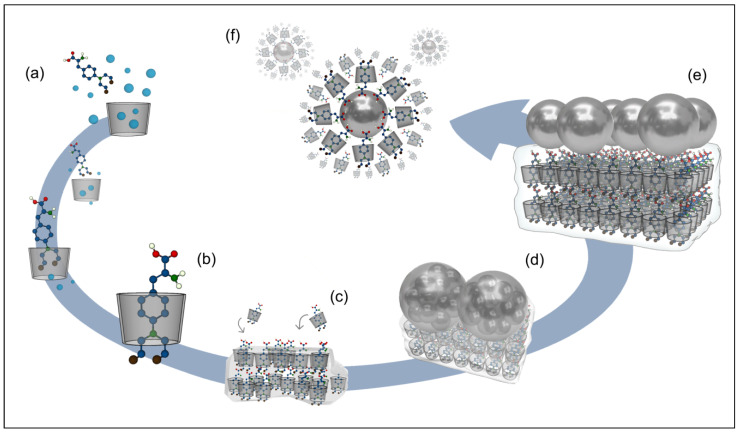
Schematic representation of nanosystem formation: inclusion process of the drug in βCD (**a**,**b**), formation of the crystals (**c**), obtaining and stabilization of AgNPs by magnetron sputtering (**d**), formation of the βCD–Mel–AgNPs crystalline system, (**e**) and solubilization process forming the βCD–Mel–AgNPs colloidal nanosystem (**f**).

**Figure 2 ijms-24-03990-f002:**
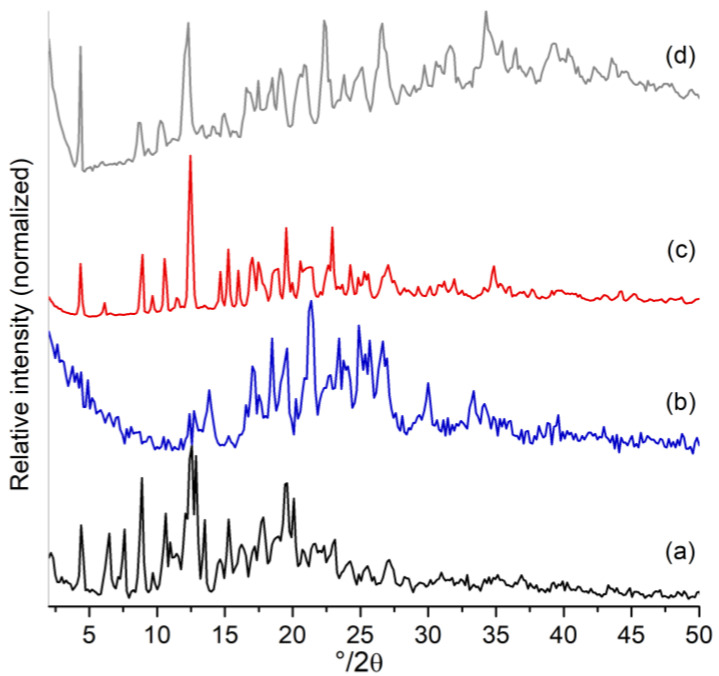
Powder diffractograms of (**a**) βCD, (**b**) Mel, (**c**) βCD–Mel, and (**d**) physical mixture of βCD and Mel.

**Figure 3 ijms-24-03990-f003:**
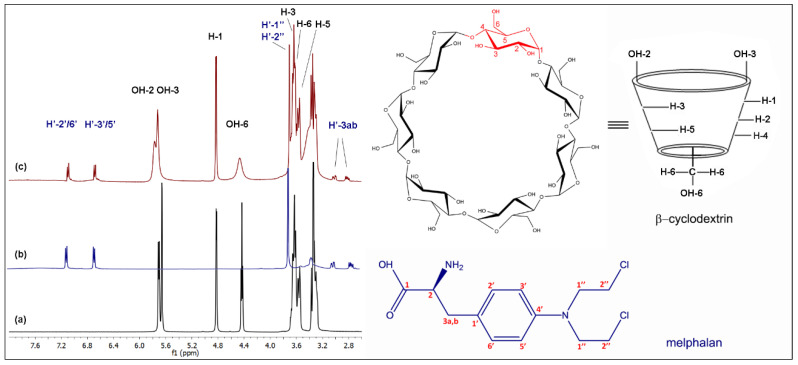
^1^H-NMR spectra of the (**a**) βCD, (**b**) Mel, and (**c**) βCD–Mel complex in DMSO-d_6_.

**Figure 4 ijms-24-03990-f004:**
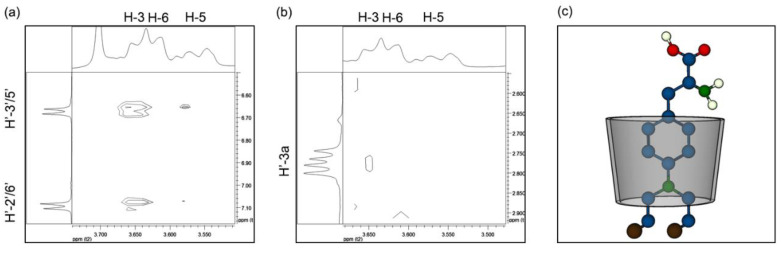
ROESY spectra of the interaction between (**a**) H′-2′/6′ and H′-3′/5′ of Mel and H-3, H-6, and H-5 of βCD; (**b**) H′-3a of Mel and H-3, H-6, and H-5 of βCD in βCD–Mel complex in DMSO-d_6_; (**c**) molecular schematic of the proposed inclusion geometry for the βCD–Mel system.

**Figure 5 ijms-24-03990-f005:**
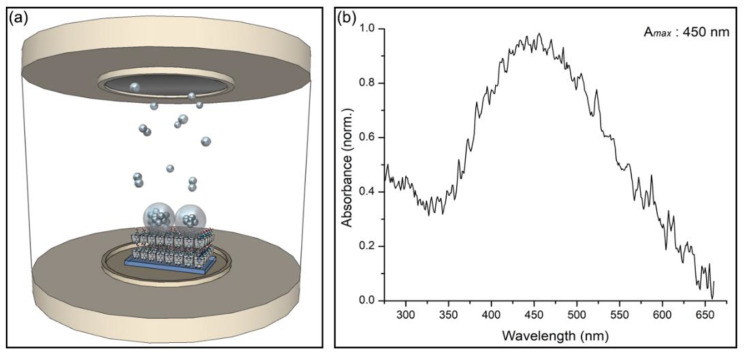
(**a**) Scheme of the release of atoms and formation of AgNPs on crystals of the βCD–Mel complex from the cathodic sputtering of a metal foil; (**b**) absorbance spectra of AgNPs deposited on βCD–Mel using magnetron sputtering.

**Figure 6 ijms-24-03990-f006:**
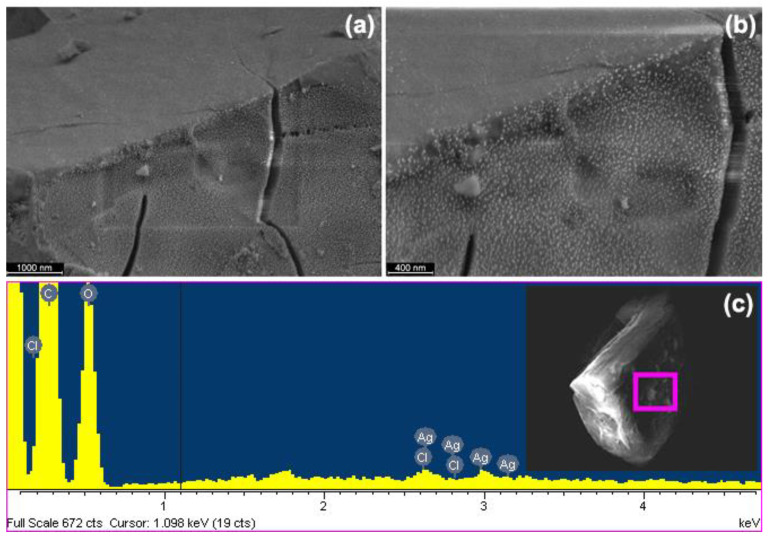
(**a**,**b**) FE-SEM micrograph of βCD–Mel with AgNPs, the time of exposure in sputtering was 32 s. (**c**) EDX spectrum taken at the marked area in the SEM micrograph at its right side of AgNPs deposited on βCD–Mel by sputtering, with elemental analysis values in the inset.

**Figure 7 ijms-24-03990-f007:**
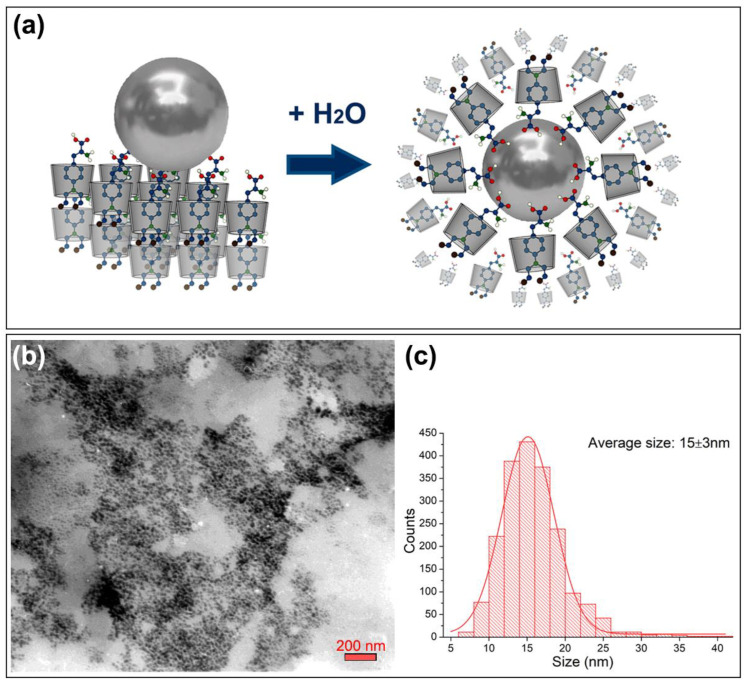
(**a**) Schematic representation of the solubilization process of AgNPs onto ꞵCD–Mel crystalline complex forming the ꞵCD–Mel–AgNPs nanosystem in colloidal solution, (**b**) TEM micrograph of βCD–Mel–AgNPs colloidal nanosystem with its respective size distribution histogram (**c**).

**Figure 8 ijms-24-03990-f008:**
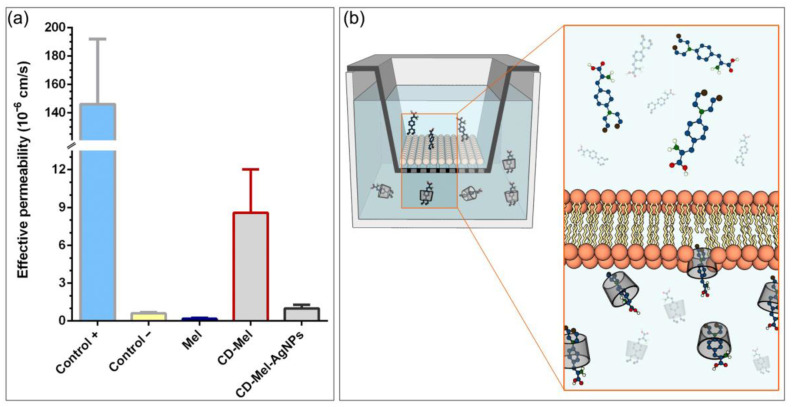
(**a**) Effective permeabilities of free Mel, Mel included in βCD and Mel in βCD–AgNPs nanosystem. Thiopental and Evans Blue solutions were used as negative and positive controls, respectively. The assays were performed at 37 °C for 24 h, using PBS as solvent, (*n* = 3); (**b**) representation of a transwell plate used in PAMPA with a scheme of permeation of the drug through an artificial membrane.

**Table 1 ijms-24-03990-t001:** Chemical shifts of the βCD–Mel complex and free species.

H of βCD	δ βCD (ppm)	δ βCD–Mel (ppm)	Δδ (ppm)	H′ of Mel	δ Mel (ppm)	δ βCD–Mel (ppm)	Δδ (ppm)
H-3	3.648	3.653	0.005	H′-1″/2″	3.699	3.702	0.003
H-5	3.555	3.570	0.015	H′-3a	2.736	2.814	0.078
H-6	3.617	3.625	0.006	H′-3b	3.017	3.012	−0.005
OH-2	5.735	5.766	0.031	H′-2′/6′	7.087	7.093	0.006
OH-3	5.680	7.719	0.039	H′-3′/5′	6.671	6.683	0.012
OH-6	4.479	4.464	−0.015				

**Table 2 ijms-24-03990-t002:** Hydrodynamic diameter, polydispersity index (PDI), zeta potential, and TEM size of βCD–Mel–AgNPs colloidal nanosystem.

System	Hydrodynamic Diameter (nm)	PDI	Zeta Potential (mV)	TEM Size (nm)
βCD–Mel–AgNPs	116 ± 63	0.40	19 ± 5	15 ± 3

## Data Availability

Not applicable.

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
