# Peer review of "Solid-State Formation of a Potential Melphalan Delivery Nanosystem Based on β-Cyclodextrin and Silver Nanoparticles"

_ijms, 2023, doi:10.3390/ijms24043990_

Round 1

Reviewer 1 Report

The presented study "Solid-state formation of a potential Melphalan delivery nanosystem based on beta-cyclodextrin and silver nanoparticles" is an example of a well-executed and well-thought-out scientific work. Motivation and novelties are also clearly defined. The article is written in a clear and understandable form.

A few typographical errors:

Missing spaces before citation (incorrectly "diseases[1–4]", should be "diseases [1–4]" - Line 29).
Variables should be written in italics, e.g. Ka (Line 166) etc.
"-1" values should be written with the correct negative sign.
The number "2" in HN2 should be written with a subscript (Line 200 or 241).
The entry "Ag1+" is not allowed. Can be "Ag+" or "AgI", or "Ag(I)" (Line 255).

Conclusions (and Abstract too) sounds very simple and modest. It would be better to highlight the achieved goals more concretely.

Reviewer 2 Report

The authors present the inclusion of Melphalan (Mel) in β-cyclodextrin (βCD) to form Mel-βCD crystals, and deposited  silver nanoparticles (AgNPs) to form βCD-Mel-AgNPs . They demonstrate the 1:1 stoichiometry of the Mel-bCD complex with a loading capacity of 27%, an association constant of 625 M-1, a degree of solubilization of 0.034, and the partial inclusion of Mel into βCD. The AgNPs, with an average size of 15±3 nm, are stabilized through NH2 and COOH groups of Mel crystals. The permeability of Mel increased using βCD. The  βCD-Mel-AgNPs nanosystem is not demonstrated, only the deposit of Ag-NPS on the βCD-Mel surface. Therefore, the  statement "forming the βCD-Mel-AgNPs nanosystem" should be modified in line 18 and wherever it appears.

The work is well performed, therefore I consider appropriate for publication in this special issue.

 Minor changes:

Line 110: “at 2θ angle of approximately 12°.” instead of “ at an angle of approximately 12° 2θ.”

Reviewer 3 Report

The authors have developed a nano complex comprising anticancer drug melphalan, β-cyclodextrin (β-CD), and silver nanoparticles (AgNPs) for enhancing physicochemical properties and anticancer activity of melphalan which is a very interesting topic. However, some important points are missing in the manuscript:

1. Various methodologies have been adopted in this research article and references are required in various sections like Synthesis of the βCD-Mel complex, Formation of silver nanoparticles, and UV-VIS spectroscopy in the solid state.

2. Quantification of melphalan by UV-VIS spectroscopy is somewhat doubtful as a more accurate finding could be obtained by authors exploring a validated HPLC method. I suggest authors to provide full data of validation of the UV-VIS spectroscopy method used for the quantification of melphalan in this research.

3. A very important aspect missing in this research is to check whether this nano complex developed is effective on some cancer cell lines or not. I suggest the authors to perform anticancer activity evaluation of this developed nano complex in-vitro against some anticancer cell lines.

4. In the case of silver nanoparticles (AgNPs) authors have not provided the important Zeta potential and Polydispersity index (PDI) data. These two are very important parameters of a nanoformulation. I request authors to provide this data. I request authors to provide high-resolution TEM images of AgNPs deposited on βCD-Mel.

5. In the section “Parallel artificial membrane permeability assay (PAMPA)” the PBS (pH = 7.4) was used. The authors must justify the use of this specific pH = 7.4 buffer in the study. I suggest the authors to perform this study exploring an acidic buffer and submit the data here because the acidic microenvironment of tumors (pH 5.6 to 6.8) as the nano complex prepared is anti-cancerous.  

Reviewer 4 Report

In my opinion, this is a potentially interesting study which requires further elaboration by the authors:

1) Exactly what role do the AgNPs play here in this form? From the title I expected this to be free AgNPs with grafted CDs, which opens up for a range of delivery pathways. Which leads me to my second question:

2) What type of delivery system is targeted here, and for what application?

3) Also, given the dimensions as shown in Figure 6, the term "nanosystem" is somewhat misleading

Reviewer 5 Report

Dear authors,

I consider you paper is adequate for being published in IJMS. In the future, some experiments concerning the biocompatibility of your obtained systems...in vitro, as well as in vivo...

Round 2

Reviewer 3 Report

Author have done all corrections and paper can be accepted in current from 

Reviewer 4 Report

The authors appear to have addressed shortcomings of the previous version